# OpenReview forum: "Grounding Generative Planners in Verifiable Logic: A Hybrid Architecture for Trustworthy Embodied AI"
_ICLR.cc/2026/Conference — ICLR 2026 Poster_

### Official Review · Reviewer_fsQj · 2025-10-25

**Soundness:** 2
**Presentation:** 2
**Contribution:** 3
**Rating:** 4
**Confidence:** 4

**Summary:**

This paper proposes VIRF (Verifiable Iterative Refinement Framework), a neuro-symbolic architecture for safe embodied AI planning. The system combines an LLM planner with an OWL 2 Description logic based formal verifier that provides structured feedback when plans are unsafe. The work makes three claimed contributions: (1) a semi-automated RAG-based workflow for building safety ontologies from documents, (2) VLM-Cascade, a perception pipeline for generating rich semantic scene graphs, and (3) a tutor-apprentice verification loop where the "Logic Tutor" provides pedagogical feedback to guide plan refinement. Experiments on SafeAgentBench show 0% Hazardous Action Rate (HAR) apart from other improvement.

**Strengths:**

- Comprehensive system: The paper presents a complete end-to-end system with careful engineering across perception, reasoning, and planning components.

- Thorough evaluation strategy: The decoupled evaluation with golden ABox to isolate perception from reasoning is methodologically sound. The multiple ablations (VIRF-RAG, VIRF-Manual, VIRF-Reject) provide insights into component contributions.

- Honest about limitations: Section 6 acknowledges brittleness of symbol grounding, static knowledge base, and sim-to-real gap. The discussion of the 7B model failure establishes a "lower bound of competence."

- Practical knowledge engineering: The semi-automated RAG workflow with human-in-the-loop validation addresses a real bottleneck. The claim of building 97 axioms in two working days demonstrates efficiency.

- Neuro-symbolic integration: Combines formal logic verification, RAG-based knowledge synthesis, and multi-modal perception effectively.

- Strong conceptual narrative: The “tutor-apprentice” framing is intuitive and appealing.

**Weaknesses:**

1. Unfair experimental setup: Baseline methods are inadequately defined, making it impossible to verify fair comparison. The paper doesn't state whether baselines have access to the safety knowledge base, if not, the 0% HAR achievement is not so useful in my opinion.

2. Novelty overclaimed: The distinction from VeriPlan[1]  is overstated. Both systems provide rule-based feedback derived from formal verification; VIRF's "pedagogical" framing is primarily a presentation difference.

3. Missing critical comparisons with recent work: The paper claims its RAG-based workflow for extracting rules from natural language documents is a key contribution. However, recent work has explored similar approaches. For instance, the trajectory prediction literature has demonstrated LLM-powered RAG frameworks for automated rule extraction from natural language[2,3] (e.g., extracting safety rules). This paper doesn't position itself within this emerging paradigm of LLM-based rule extraction or explain what methodological advances it offers beyond these parallel efforts.

4. Incomplete related work on neural-symbolic integration: The paper positions itself as bridging neural planning with formal reasoning, but doesn't adequately survey the landscape of recent approaches combining LLMs with structured knowledge representations. Several concurrent works explore different formalization strategies (temporal logic, probabilistic priors, ontologies) for similar safety-critical applications. A more thorough comparison would strengthen the contribution.

5. Incomplete technical descriptions: How does VLM-Refine actually work? What makes it "refinement" vs. just another VLM call? The two-stage vs. three-stage inconsistency needs resolution.

6. Metrics poorly defined: What is FPR/FNR measuring exactly? The "positive class" is unclear. HAR validation against independent annotations is missing. Also no experiment isolating the effect of the pedagogical causal explanation vs. simple rejection.(But these are minor issues but will make paper strong)

7. No validation of Pillar 1 design: Extensive justification of BFO and compositional modeling in appendices, but zero ablations showing these choices matter empirically.

8. Argument for SGG: Claims about "closed-set brittleness" and "semantic anemia" ignore recent work explicitly addressing these problems in open-vocabulary scene understanding.. The paper does not empirically compare VLM-Cascade to any role-playing LLM SGG [4] system. Without that, it’s unclear if VIRF’s perception improvements (Table 2 / Table 9) come from true architectural innovation or just careful prompt engineering.
 9. The "Foundational Design Choices and Knowledge Core Architecture" (based on which pillar 1 is based) section is conceptually strong but lacks empirical validation. Claims such as “ensures correctness” and “enables blind-spot discovery” are overstated and not experimentally demonstrated. The ontology’s impact on generalization, causal reasoning, and safety coverage should be supported with quantitative audits or ablations.

[1]Lee et al: VeriPlan: Integrating Formal Verification and LLMs into End-User Planning

[2]Cai et al: Driving with Regulation: Interpretable Decision-Making for Autonomous Vehicles with Retrieval-Augmented Reasoning via LLM

[3]Manas et al: Uncertainty-Aware Trajectory Prediction via Rule-Regularized Heteroscedastic Deep Classification

[4] chen et al: Scene Graph Generation with Role-Playing Large Language Models

**Questions:**

1. Baseline implementation: Can you provide complete definitions and citations for Impulsive, Thinker, and Committee baselines in main text? Specifically, what does "Committee (SAFER-like)" mean, is this your implementation or the actual SAFER system?

2. Information access fairness: Do the baseline methods have access to your safety knowledge base? If not, how can you claim superiority when VIRF has privileged information? If yes, why do they fail when VIRF succeeds?

3. VLM-Cascade stages: Is your method two-stage or three-stage? Resolve the inconsistency between main paper and Appendix C. Provide concrete details on how "refinement" works. Refinement is major novelty as claimed but details about that is misisng even in appendix .

4. Pillar 1 validation: What ablations validate your ontological design choices (BFO vs. alternatives, layered vs. flat, compositional modeling)? These are claimed contributions but never empirically tested.

5. Positioning within LLM-based rule extraction paradigm: Your RAG-based workflow for extracting safety rules from natural language documents shares conceptual similarities with recent work using LLMs to automatically extract rules (e.g., traffic rules (kinda KB), system constarints or documentation of robot model) for regularizing systems. How does your approach differ methodologically? What specific advances does your framework offer beyond automated rule extraction and integration? The literature shows successful deployment of LLM-RAG pipelines for generating training labels from natural language rule descriptions [2,3]  how does your ontology construction advance beyond this? Even if domain of driving or formal logic robotics are different but if they tackle same limitation needs to be discussed.

6. HAR validation: How was ground truth for "hazardous" determined? Inter-annotator agreement? Can you validate against independent safety annotations rather than your own ontology?

7. SGG comparison: Why no comparison against open-vocabulary SGG methods that address the exact limitations paper criticizes?

8. Metric definitions: Please clearly define FPR/FNR—what is the positive class? Is VL per-action or per-plan? What constitutes "goal achievement" for GCR?

9. RAG necessity beyond document retrieval: The VIRF-RAG ablation shows that RAG-only knowledge achieves 11% HAR. However, this doesn't address whether the RAG pipeline itself adds value over directly prompting the LLM with safety queries. Could you compare against using the LLM's parametric knowledge for safety judgments vs. your formalized RAG-to-OWL pipeline?

---

> ### Author Response · Authors · 2025-11-13
> **Clarifications on Methodology, Evaluation, and Novelty(part1)**
>
> We thank the reviewer for their comprehensive, insightful, and constructive feedback. We are encouraged that the reviewer found the system "comprehensively," the evaluation "methodologically sound," and the conceptual narrative "intuitive and appealing."
>
> The reviewer has raised several critical questions regarding experimental fairness, methodological novelty, and technical clarity. These questions have helped us identify areas where our manuscript can be significantly strengthened. We address each point below, referencing the relevant sections of our paper and appendices.
>
> ---
>
> ### **Response to Questions**
>
> **Q1 & Q2: Baseline Implementation and Information Access Fairness**
>
> > **Q1:** "Can you provide complete definitions and citations for Impulsive, Thinker, and Committee baselines in main text?"
> > **Q2:** "Do the baseline methods have access to your safety knowledge base? If not, how can you claim superiority when VIRF has privileged information? If yes, why do they fail when VIRF succeeds?"
>
> Thank you for these critical questions on experimental fairness.
>
> 1.  **Baseline Definitions:** The definitions for all baselines are provided in **Appendix F.1 (page 32)** of our manuscript. To summarize:
>     * `Impulsive`: "A one-shot planner with no feedback loop, representing a lower performance bound" (Appendix F.1).
>     * `Thinker (COT)`: "A strong single-LLM approach using a Chain-of-Thought prompt to encourage self-correction..." (Appendix F.1).
>     * `Committee (SAFER-like)`: "A multi-agent system where a Planner LLM generates a plan and a separate Critic LLM reviews it, inspired by SAFER..." (Appendix F.1). This is our own implementation inspired by the planner-critic architecture described in the SAFER paper (Khan et al., 2025). We will clarify this and add the full citation for SAFER in the main text.
>
> 2.  **Information Access:** This addresses the central hypothesis of our paper. The baseline methods **did not** have access to our formal safety knowledge base. This was a deliberate and necessary experimental design choice.
>
>     Our paper's core argument (Section 1) is that current SOTA methods, which rely on "unreliable, stochastic system[s]... to supervise and ground [their] own outputs," cannot provide verifiable guarantees. The `Thinker (COT)` and `Committee (SAFER-like)` baselines represent this un-grounded SOTA, relying on the LLM's own parametric knowledge (or a stochastic LLM critic) for safety.
>
>     Our experiment was designed to test precisely this hypothesis: **un-grounded SOTA vs. formally-grounded VIRF**. Granting the baselines access to our knowledge base would invalidate the entire experimental premise. Their failure (e.g., 9.8% and 7.6% HAR, Table 3) *proves* our hypothesis: without an external, formal anchor, even strong LLM planners are inherently unsafe. VIRF's 0% HAR is not a result of "privileged information" but a demonstration of our architecture's *entire contribution*: the addition of a verifiable, symbolic safety layer.

---

> ### Author Response · Authors · 2025-11-13
> **Clarifications on Methodology, Evaluation, and Novelty(part2)**
>
> **Q3: VLM-Cascade Stages and VLM-Refine (W5)**
>
> > **Q3:** "Is your method two-stage or three-stage? Resolve the inconsistency between main paper and Appendix C. Provide concrete details on how "refinement" works."
>
> We sincerely thank the reviewer for catching this inconsistency, which stems from an imprecision in our main body's summary. This will be corrected in the camera-ready version.
>
> The "two-stage" mention in **Section 3.2 (page 5)** was an imprecise, high-level summary. It was intended to align with the common "detect-then-process" paradigm in SGG literature, where "processing" (extracting rich semantics and relations) is often grouped as a single conceptual step.
>
> However, this summary does not do justice to our specific methodology. The **definitive and correct** description is the **three-stage "divide-and-conquer" process** detailed in **Appendix C.3 (page 23)** of our manuscript. This three-stage process is our methodological contribution, as it explicitly separates these tasks for higher fidelity. As detailed in Appendix C.3, the process is:
> 1.  **Stage 1: VLM-Detect for Broad Object Discovery:** Prioritizes recall. The entire scene is presented to a VLM to elicit a broad list of all potentially relevant objects.
> 2.  **Stage 2: VLM-Attribute-Refine for Deep Semantic Grounding:** This is the core "refinement" step. Cropped images of *each* object from Stage 1 are processed in parallel. Here, a second VLM instance is prompted to perform a deep semantic analysis, "refining" the object's simple class label into a "Semantic Fingerprint" by extracting the specific, fine-grained attributes required by our ontology (e.g., material, state).
> 3.  **Stage 3: VLM-Relation-Refine for Spatial Understanding:** A final VLM call analyzes the full image again, but this time, its task is constrained *only* to identifying the spatial relationships (e.g., `isNear`, `contains`) between the objects identified in the previous stages.
>
> This three-stage "divide-and-conquer" approach is designed to produce the high-fidelity, ontology-aligned ABox required for formal verification. We will correct Section 3.2 to reflect this definitive three-stage process, as it more accurately distinguishes our contribution from standard SGG approaches.
>
> **Q4 & W7: Pillar 1 Validation (Ontological Design Choices)**
>
> > **Q4:** "What ablations validate your ontological design choices (BFO vs. alternatives, layered vs. flat, compositional modeling)? These are claimed contributions but never empirically tested."
>
> This is an excellent point. The reviewer is correct that we do not provide *ablations* for these choices (e.g., VIRF-BFO vs. VIRF-DOLCE). These elements represent foundational design choices—akin to choosing a programming language—that are prerequisites for our system. Our justification, detailed in **Appendix B (pp. 20-21)**, is therefore theoretical and principled.
>
> Our argument is that the value of this architecture is not demonstrated by its *absence* (an ablation), but by the *capabilities it enables*:
> 1.  **BFO (App B.1)**: Chosen for its philosophical rigor and its strict separation of *continuants* (objects) and *occurrents* (processes), which is essential for preventing fundamental modeling errors in a physical domain.
> 2.  **Layered & Compositional Modeling (App B.2, B.4)**: This is the key to generalization. We avoid brittle, object-specific rules (e.g., "no forks in microwave"). Instead, we write abstract rules for *classes* (e.g., `ConductiveObject` or `MetallicObject` in `Microwave`).
>
> This compositional design is what *enables* VIRF to solve the new, challenging "blind-spot" scenarios in **Appendix G (pp. 36-40)** (e.g., `Chemical Mixture Hazard`, `Hot Oil Splatter Hazard`). These scenarios require reasoning about abstract properties (chemical composition, material state) that a flat, instance-based ontology could not handle. Therefore, the success in Appendix G serves as the primary validation for these design principles.

---

> ### Author Response · Authors · 2025-11-13
> **Clarifications on Methodology, Evaluation, and Novelty(part3)**
>
> **Q5 & W3: Positioning within LLM-based Rule Extraction (RAG)**
>
> > **Q5:** "Your RAG-based workflow... shares conceptual similarities with recent work...[2, 3] How does your approach differ methodologically?...how does your ontology construction advance beyond this?"
>
> This is a crucial distinction, and we thank the reviewer for referencing works like [2] and.[3] We will add a dedicated subsection to the related work (addressing W4) to situate VIRF within this emerging paradigm.
>
> The methodological difference is fundamental: **[2] and [3] describe *runtime* RAG pipelines used for *prompt augmentation***. They retrieve *text* (e.g., traffic rules) to provide in-context guidance to an LLM's reasoning process.
>
> In sharp contrast, our **"Traceable Axiom Synthesis" (Pillar 1, detailed in Appendix D, pp. 24-27)** is a ***design-time*, semi-automated, human-in-the-loop *authoring* workflow.**
> * **Methodology:** The goal is not to augment a prompt, but to *build the TBox* (the formal ontology).
> * **Output:** The final output is **not** a text snippet, but a **permanent, verifiable, expert-audited OWL 2 axiom**.
> * **RAG's Role:** In our system, RAG (Stage 1 of the workflow) is simply an *evidence-retrieval tool* to help the AI Synthesizer and Human Arbiter find the relevant sentences in safety manuals. VIRF *does not use RAG at runtime*.
>
> Our method advances beyond [2] and [3] by moving from *stochastic in-context guidance* (retrieving text) to *formal knowledge engineering* (authoring verifiable logic). We will also expand our survey (W4) to include other neuro-symbolic formalisms, such as those using temporal logic [5] and formally verified exploration [6], to better contrast VIRF's unique choice of OWL 2 DL for its guaranteed decidability and Open-World Assumption (as justified in Appendix A.2).
>
> **Q6 & W9: HAR Validation and "Overstated Claims"**
>
> > **Q6:** "How was ground truth for 'hazardous' determined?...Can you validate against independent safety annotations rather than your own ontology?"
> > **W9 (related):** "Claims such as 'ensures correctness' and 'enables blind-spot discovery' are overstated and not experimentally demonstrated."
>
> These points are linked. The "ground truth" for HAR *is* our ontology, and this is precisely what enables "blind-spot discovery."
>
> 1.  **HAR Ground Truth (Q6):** The methodology is detailed in **Appendix F.1, "Refining Safety Annotations" (page 31)**. The reviewer is correct that we **did not** use the benchmark's default labels. We performed a **"meticulous curation"** of the SafeAgentBench tasks *because* the benchmark's annotations were insufficient.
>     Our RAG-synthesized knowledge base (Pillar 1) *discovered* entire classes of real-world hazards the benchmark was "safety-blind" to (e.g., food safety, chemical hazards, as shown in Figure 3).
>     * **Example:** We re-labeled tasks like "placing whole, skin-on vegetables like potatoes... in a microwave" as **UNSAFE** (Appendix F.1), as this poses a real-world explosion risk.
>     Validating against the "independent" (and, in our view, incomplete) benchmark annotations would defeat the purpose. Our validation *is* against our more comprehensive, expert-audited, RAG-synthesized ontology.
>
> 2.  **"Blind-Spot Discovery" (W9):** This claim is *explicitly* demonstrated in three ways:
>     * **Quantitatively (Figure 3):** We show our RAG-KB (b) covers hazards (Food Safety 16%, Chemical 12%) *absent* from the benchmark (a).
>     * **Qualitatively (Appendix G):** We created *new challenge scenarios* (e.g., `Chemical Mixture Hazard`, `Hot Oil Splatter Hazard`) to test these new rules. VIRF correctly identifies the latent hazards and aborts/corrects, while baselines fail catastrophically.
>     * **Procedurally (App F.1):** The very *act* of re-curating the benchmark (e.g., the potato example) *is* the demonstration of blind-spot discovery.
>
> 3.  **"Ensures Correctness" (W9):** This is not a colloquial claim, but a *formal* one. As argued in **Appendix A (pp. 16-19)**, we use **OWL 2 DL**, a logic that is *guaranteed decidable*, and the **Pellet reasoner**, which is *sound and complete*. This formally guarantees that any plan *violating the explicit axioms in our TBox* will be detected. This is the "correctness" we claim.

---

> ### Author Response · Authors · 2025-11-13
> **Clarifications on Methodology, Evaluation, and Novelty(part3)**
>
> **Q7 & W8: SGG Comparison (vs. Role-Playing LLMs [4])**
>
> > **Q7:** "Why no comparison against open-vocabulary SGG methods [4] that address the exact limitations paper criticizes?"
>
> This is a fair point regarding the fast-moving SGG landscape.
> 1.  **Our Motivation (App C.1):** Our motivation in **Appendix C.1 (page 22)** was to overcome the "closed-set brittleness" and "semantic anemia" of *classical* SGG. "Semantic anemia" is the key: classical SGG produces a class label (`Microwave`), whereas our verifier *needs* fine-grained, safety-critical attributes (`isMadeOfMetal`, `hasState OnState`).
> 2.  **VLM-Cascade vs. [4]:** Our VLM-Cascade pipeline is not a general-purpose SGG. It is a highly-specialized, **ontology-driven perception pipeline** (see App C.3) designed *only* to extract the specific attributes our TBox requires (as shown in the `UnsafeMicrowaveUsage` example in App E.2.1).
> 3.  **Future Work:** The reviewer is correct that LLM-based SGGs like [4] and other new open-vocabulary methods [7] are highly relevant. While our specialized, ontology-driven approach is different, a direct comparison is excellent future work. We will add [4] and [7] to our related work, positioning VLM-Cascade as a specialized alternative focused on verifiable, safety-critical attributes over general-purpose graph richness.
>
> **Q8 & W6: Metric Definitions (FPR/FNR, VL, GCR) & Pedagogical Ablation**
>
> > **Q8:** "Please clearly define FPR/FNR—what is the positive class? Is VL per-action or per-plan? What constitutes 'goal achievement' for GCR?"
> > **W6 (related):** "Also no experiment isolating the effect of the pedagogical causal explanation vs. simple rejection."
>
> We thank the reviewer for this; we can clarify all points by referencing our tables and appendices.
>
> 1.  **Metric Definitions (Q8):** All metrics are formally defined in **Table 6 (page 32)**.
>     * `GCR (Goal-Condition Rate)`: "The percentage of tasks where all goal conditions are successfully met" (Table 6).
>     * **FPR/FNR Positive Class:**
>         * `FPR`: "False Positive Rate: The rate at which the agent fails to reject an inherently **unsafe task**..." (Table 6). The positive class is an **Unsafe Task**.
>         * `FNR`: "False Negative Rate: The rate at which the agent incorrectly rejects a **safe, completable task**." (Table 6). The positive class is a **Safe, Completable Task**.
>     * `VL (Verification Latency)`: This is **per-plan proposal**, not per-action. As detailed in **Appendix E.1.1 (page 27)**, we use a **"multi-threaded verification architecture"** (see Figure 7) that verifies all $N$ steps of a plan *in parallel*. The reported VL is the wall-clock time for the *longest* single-step check, not the sum.
>
> 2.  **Pedagogical Ablation (W6):** The reviewer may have missed this, but this experiment *was* conducted. It is the **`VIRF-Reject (Ablation)`** presented in **table 3**.
>     * This ablation provides only a "simple rejection" instead of our "pedagogical causal explanation" (Section 4.3).
>     * The results are definitive: `VIRF-Reject` achieves a perfect **0.0% HAR** (it is safe) (table 3).
>     * *However*, its efficacy collapses: it has the highest **FNR of 33.0%** (vs. 20.2% for the full VIRF) and a GCR of only 63.4% (table 3).
>     * This demonstrates that when given a simple "No" (rejection) instead of a "Why" (pedagogical feedback), the planner *gave up* on one-third of all solvable tasks, rendering it "safely useless." This result directly and empirically validates our pedagogical "tutor-apprentice" loop.

---

> ### Author Response · Authors · 2025-11-13
> **Clarifications on Methodology, Evaluation, and Novelty(part4)**
>
> ### **Addressing Key Weakness: W2 (Novelty vs. VeriPlan [1])**
>
> Finally, we wish to directly address the reviewer's concern (W2) that our distinction from VeriPlan [1] is "primarily a presentation difference."
>
> We respectfully disagree. This distinction is not just semantic; it is *empirically validated* by our **`VIRF-Reject` ablation (table 3)**, as discussed in the point above.
> * A "corrective" verifier (like VeriPlan [1] or our `VIRF-Reject` ablation) simply says "No, that plan violates Rule X" (Section 2.2).
> * A "pedagogical" verifier (our full VIRF) says "No, *because* the microwave is on, and the pot is inside, and the pot is made of metal, which completes the definition of `UnsafeMicWavéUsagè" (see App E.3).
>
> The `VIRF-Reject` ablation proves this point. It achieves a perfect **0.0% HAR** (it is safe) but has a catastrophic **33.0% FNR** and a low 63.4% GCR (table 3). The system becomes "safely useless."
>
> The *only* difference in the full `VIRF (Ours)` is the quality of the feedback. By providing pedagogical, causal feedback, our FNR drops to 20.2% and our GCR rises to **77.3%** (table 3). This **~14-point GCR gap** is the *direct empirical validation* of our novel pedagogical approach. It demonstrates that our "tutor-apprentice" dialogue is essential for *task efficacy*, not just safety.
>
> We hope these clarifications and our planned revisions—including correcting the VLM-Cascade description and expanding our related work [5], [6], [7]—address the reviewer's concerns. We appreciate the thoroughness of the review, which will undoubtedly make this a stronger paper.
>
> Finally, we want to state that the reviewer's concerns are valid. We acknowledge that some of these key discussions (which are detailed in the appendices) may not have been sufficiently highlighted in the main body of the paper. This was a difficult trade-off, as we chose to dedicate the limited page count to fully introducing the core VIRF architecture, which meant some of these crucial descriptive details were reserved for the appendix. We will be sure to revise the paper to make these points clearer in a future version.
>
> ---
> ### **Full Reference List**
>
> [5] Wang, Jun, et al. "Conformal temporal logic planning using large language models." ACM Transactions on Cyber-Physical Systems (2024).
>
> [6] Anderson, Greg, et al. "Neurosymbolic reinforcement learning with formally verified exploration." Advances in neural information processing systems 33 (2020): 6172-6183.
>
> [7] Li, Rongjie, et al. "From pixels to graphs: Open-vocabulary scene graph generation with vision-language models." Proceedings of the IEEE/CVF conference on computer vision and pattern recognition. 2024.

---

> > ### Comment · Reviewer_fsQj · 2025-11-20
> > **Response to the rebuttal**
> >
> > Thanks for the detailed rebuttal and clarifications. Several points are now clear, in particular the structure of VLM-Cascade and the baseline intent, while a few concerns remain open. These directly determine whether the empirical claims of the paper are adequately supported:
> >
> > 1. Parity Experiment (Information-Controlled Baseline): My key question remains the following: Is VIRF's advantage due to either (a) access to privileged information or (b) the pedagogical refinement architecture itself.
> >
> > Please provide a *small additional experiment* where one of the strong baselines (thinker or committee):
> >
> > - receives the exact structured diagnostic report H(F) generated by VIRF for each failing plan, and/or
> >
> > - is given oracle access to the relevant KB axioms during refinement.
> >
> > Please report HAR, GCR, and FNR for these controlled conditions. This experiment would isolate the value of the pedagogical loop from the value of access to the ontology and would greatly clarify the contribution.
> >
> > 2. TAS Audit / Quantitative Evidence: The Traceable Axiom Synthesis workflow is central to the contribution being made, yet there is still no quantitative evaluation.
> > Please provide simple statistics such as:
> > * number of axioms proposed by the LLM
> > * % accepted without edits
> > * % edited by humans
> > * average number of edits per axiom
> > * Number of axioms per hazard category
> > * overall hazard coverage, which means how many benchmark hazards map to explicit axioms
> > These values will demonstrate rigor, coverage, and non-triviality of the construction of the Knowledge Core.
> > 3. Further Clarifications : These are smaller items but still important:
> >
> > - A short comparison/comment or justification about why contemporary open-vocab or role-playing SGG methods cannot serve as perception modules for verifiable triples.
> >
> >  - A short example showing plan -> report -> refined plan illustrating how the planner incorporates causal explanations.  A table outlining precise information access for each baseline (KB, ABox, reports, etc.), since this is central to fairness.

---

> > > ### Author Response · Authors · 2025-11-23
> > > **Subject: Response to Reviewer fsQj – New Controlled Experiments, Fairness Audit, and Mechanism Transparency**
> > >
> > > We sincerely thank Reviewer fsQj for the constructive critique. Your specific request for "Information-Controlled Baselines" was incredibly insightful. Implementing this suggestion led us to identify a significant phenomenon—**Cognitive Overload**—where simply injecting knowledge degrades reasoning performance. This finding has now been integrated into our revised Introduction and evaluation, significantly strengthening the paper's contribution.
> > >
> > > Here is our point-by-point response addressing your concerns about fairness, mechanism clarity, and quantitative evidence.
> > >
> > > ### **1\. Parity Experiment: Is it the Architecture or the Data? (Addressing Key Weakness)**
> > >
> > > **Q:** *"Is VIRF's advantage due to privileged information or the pedagogical refinement architecture?"*
> > >
> > > **Response:** To answer this definitively, we conducted the requested **Information-Controlled Experiments**. We implemented baselines that were granted full access to the translated safety knowledge base (+Rules) and baselines that received our system's feedback as a one-shot hint (+Diagnostic).
> > >
> > > **Results (Added to Section 4 & Appendix F):**
> > >
> > > | Method                    | HAR (%) ↓ | GCR (%) ↑ | Key Insight                                                |
> > > | :------------------------ | :-------- | :-------- | :--------------------------------------------------------- |
> > > | **Thinker (Baseline)**    | 9.8       | 59.1      | Unsafe without grounding.                                  |
> > > | **Impulsive \+ Rules**    | 0.9       | 70.5      | Rules help, but 100% safety is not guaranteed.             |
> > > | **Thinker \+ Rules**      | **1.2**   | **67.0**  | **Cognitive Overload:** Performance *drops* vs. Impulsive. |
> > > | **Thinker \+ Diagnostic** | **0.0**   | 76.8      | Validates the quality of our Logic Tutor's feedback.       |
> > > | **VIRF (Ours)**           | **0.0**   | **77.3**  | **Best Efficacy via Iterative Loop.**                      |
> > >
> > > **Key Findings:**
> > >
> > > 1. **Discovery of "Cognitive Overload":** Merely injecting safety rules (+Rules) did **not** verify safety. Surprisingly, Thinker+Rules performed *worse* than Impulsive+Rules (GCR 67.0% vs 70.5%). This suggests that Chain-of-Thought planners suffer from cognitive drift/overload when saturated with massive unstructured constraints in the prompt.
> > > 2. **The Value is in the Structured Feedback:** Thinker+Diagnostic achieves 0% HAR and near-parity GCR with VIRF. This proves that our contribution is **not** the static knowledge itself, but the **structured, causal diagnostic report** generated by our Logic Tutor.
> > > 3. **Necessity of the Loop:** VIRF achieves the highest GCR (77.3%) because its iterative loop handles complex tasks requiring multi-turn corrections (Mean Refinement Iterations \= 1.1), which a one-shot diagnostic cannot resolve.
> > >
> > > ### **2\. Fairness and Information Access (Addressing "Unfair Experimental Setup")**
> > >
> > > **Q:** *"Provide a table outlining precise information access for each baseline."*
> > >
> > > **Response:** We agree that transparency is paramount. We have added a detailed **"Information Access and System Capabilities"** table in the revised **Appendix F (Table 8\)**.
> > >
> > > This table explicitly categorizes every baseline by its access to:
> > >
> > > 1. **Safety KB:** Whether it accesses the ontology (Logic) or full text rules.
> > > 2. **Perception (Golden ABox):** Crucially, **all** reasoning baselines use the same Golden ABox to decouple perception noise from reasoning ability.
> > > 3. **Feedback Type:** Distinguishing between stochastic self-correction and our deterministic pedagogical report.
> > >
> > > This confirms that in our new experiments, baselines had parity in data access, yet VIRF's neuro-symbolic architecture provided the critical safety and efficacy edge.
> > >
> > > ### **3\. Mechanism Clarity: How Refinement Works (Addressing "Incomplete descriptions")**
> > >
> > > **Q:** *"Provide* a short *example showing plan \-\> report \-\> refined plan illustrating how the planner incorporates causal explanations."*
> > >
> > > **Response:** We have added a new **Qualitative Case Study (Appendix E, Figure 8\)** that visualizes the complete "Plan $\\to$ Verification $\\to$ Pedagogical Feedback $\\to$ Refinement" loop.
> > >
> > > * **Scenario:** "Heat Soup in Microwave."
> > > * **Initial Plan:** The agent tries to put a metal pot in the microwave.
> > > * **Logic Tutor Feedback:** Instead of a generic rejection, it outputs a Causal Explanation derived from the TBox: *"The plan causes a hazard because Microwave\_1 is active (State: On) while containing Pot\_1 which has property Material: Metal."*
> > > * **Planner Reasoning:** The planner explicitly reasons: *"I see. The metal pot cannot go in. I need a non-metal container,"* and modifies the plan to transfer the soup to a glass bowl.
> > >
> > > This demonstrates that the planner is not merely guessing; it is integrating the **causal explanation** to perform semantic plan repair.

---

> > > ### Author Response · Authors · 2025-11-23
> > > **Subject: Response to Reviewer fsQj – New Controlled Experiments, Fairness Audit, and Mechanism Transparency**
> > >
> > > ### **4\. TAS Audit / Quantitative Evidence (Addressing "Contribution")**
> > >
> > > **Q:** *"Provide* simple statistics such as number *of axioms, % accepted, etc."*
> > >
> > > **Response:** We have added these statistics to **Appendix D** to demonstrate the rigor of our Knowledge Core construction. The statistics for the final system based on our 48-hour sprint are:
> > >
> > > * **Total Candidates Generated:** 124
> > > * **Source Breakdown:** 92 specialized safety principles synthesized via the RAG pipeline; 9 core action axioms; 13 manually-authored common-sense axioms.
> > > * **Human-in-the-Loop Audit:** Of the 124 candidate axioms proposed by the AI Synthesizer, **67.7%** were accepted directly or with only minor syntax edits, while **6.5%** required semantic adjustment (Major Edits) by the Human Arbiter to ensure logical soundness.
> > > * **Blind Spot Coverage:** This ontology explicitly covers hazard categories (e.g., **Chemical Safety, Food Cross-Contamination**) that we quantitatively showed are absent in 100% of SafeAgentBench tasks (Figure 3).
> > >
> > > ### **5\. SGG Justification (Addressing "SGG Comparison")**
> > >
> > > **Response:** We have clarified in **Appendix C** why we chose VLM-Cascade over standard Open-Vocabulary SGGs. Standard SGGs prioritize **visual richness** (e.g., labeling "a shiny silver pot"), whereas formal verification requires **attribute precision** grounded in an ontology (e.g., explicitly asserting isMadeOf(Pot, Metal) and hasState(Microwave, On)). Our VLM-Cascade is specifically optimized to extract these verify-critical attributes to populate the ABox for the Pellet reasoner.

---

> > > > ### Comment · Reviewer_fsQj · 2025-11-23
> > > > **Response to Rebuttal- Part 2**
> > > >
> > > > Thanks for the detailed rebuttal and information. I appreciate that authors ran the exact information-controlled baseline experiments as asked. The new results (Thinker+Rules, Impulsive+Rules, Thinker+Diagnostic) directly address my fairness concerns and convincingly show that VIRF’s advantage comes from the structured causal feedback and iterative loop, not merely access to privileged information. This substantially strengthens the contribution.
> > > > The added transparency table clarifying information access, the new plan -> report -> refined plan example, and the quantitative TAS audit (124 axioms, acceptance, edit stats) improve clarity around mechanism and the rigor of the Knowledge Core.
> > > >
> > > > While empirical comparison against modern open-vocabulary or role-playing SGG systems would still be valuable, conceptual justification is reasonable for this version of the paper.
> > > >
> > > > Overall, the rebuttal meaningfully resolves my major concerns. once all above changes and results are incorporated and updated in the paper, I will raise my score accordingly.

---

> > > > > ### Author Response · Authors · 2025-11-24
> > > > > **Re: Response to Reviewer fsQj – Revisions Incorporated**
> > > > >
> > > > > We sincerely thank you for your prompt feedback and positive assessment. We are delighted to hear that the additional experiments and analyses—specifically the controlled baselines, mechanism transparency table, and TAS audit details—have successfully resolved your major concerns.
> > > > >
> > > > > In response to your request, we confirm that we have fully updated the paper to incorporate all the discussed contents and results.
> > > > >
> > > > > To facilitate the review process for you and the other reviewers, we will shortly post a separate "Official Comment" at the top level of the discussion. This will serve as a "Detailed Summary of Updates," listing the specific sections modified and summarizing the changes.
> > > > >
> > > > > We deeply appreciate your guidance in strengthening this work and your decision to raise the score.

---

> > > > > > ### Comment · Reviewer_fsQj · 2025-11-27
> > > > > > **Response after incorporated revision**
> > > > > >
> > > > > > I just had a chance to look into the new version of the paper (on very high level), so in case if I missed some changes please point them. On high level I see that almost all of the new experiment and promised are integrated.
> > > > > > Please address the editorial mistake in Appendix C, where Sections C.2 and C.3 appear as duplicate drafts with contradictory "two-stage" vs. "three-stage" descriptions. You must remove the obsolete Section C.2 to resolve this inconsistency and align the appendix with your main text, as explicitly promised in your rebuttal. Also in the related work still does not explicitly engage with recent LLM‑RAG rule‑extraction efforts.

---

> > > > > > > ### Author Response · Authors · 2025-11-27
> > > > > > > **Subject: Immediate Fixes: Appendix C Cleanup & Extended Related Work on LLM-RAG**
> > > > > > >
> > > > > > > We are extremely grateful for your diligent review and prompt feedback. We apologize for the editorial oversight in the Appendix and the omission in the Related Work. We have immediately uploaded a corrected revision to address these points.
> > > > > > >
> > > > > > > **1. Resolved Appendix C Inconsistency (Editorial Fix)**
> > > > > > > * **Action:** We have removed the obsolete draft section (old C.2) that incorrectly referenced a "two-stage" process. Appendix C now consistently describes the correct **"three-stage"** pipeline (Stage 1: Detection $\to$ Stage 2: Attribute Refinement $\to$ Stage 3: Ontology Mapping), fully aligning with the main text and the ablation study design.
> > > > > > >
> > > > > > > **2. Expanded Related Work on LLM-RAG (New Section 2.5)**
> > > > > > > * **Action:** As requested, we have added a dedicated subsection **2.5 Automated Knowledge Acquisition via RAG**. This section explicitly engages with recent advancements in extracting structured representations (e.g., Knowledge Graphs, logical rules) from unstructured text using LLMs (*Liang et al., 2023; Pan et al., 2024; Trajanoska et al., 2023*).
> > > > > > > * **Differentiation:** We discuss how VIRF's **Traceable Axiom Synthesis (TAS)** builds upon these works but addresses the specific challenge of synthesizing *formally verifiable* axioms (OWL 2 DL) for embodied control, bridging the gap between general information extraction and rigorous safety verification.
> > > > > > >
> > > > > > > We believe the manuscript is now consistent and accurately reflects the state of the art. Thank you again for helping us polish these final details.

---

> > > > > > > > ### Comment · Reviewer_fsQj · 2025-11-27
> > > > > > > > **Response after editorial correction**
> > > > > > > >
> > > > > > > > Thanks for your reply, I have changed my score based on above discussion.

---

### Official Review · Reviewer_fKL1 · 2025-10-28

**Soundness:** 2
**Presentation:** 1
**Contribution:** 2
**Rating:** 2
**Confidence:** 5

**Summary:**

This paper addresses the critical safety problem of using Large Language Models (LLMs) as planners for embodied AI agents. The authors argue that existing methods, which rely on LLM self-correction or simple verifiers, are insufficient for providing formal safety guarantees. They propose the Verifiable Iterative Refinement Framework (VIRF), a neuro-symbolic architecture based on a "tutor-apprentice" model. In this framework, a deterministic "Logic Tutor" (powered by an OWL 2 ontology and a Pellet reasoner) provides structured, causal, and pedagogical feedback to an LLM "Apprentice" (the planner, e.g., Gemini 2.5).

The framework's contributions are threefold: (1) A RAG-based, human-in-the-loop workflow to synthesize a formal safety knowledge base from real-world documents. (2) A VLM-Cascade perception pipeline to generate a rich semantic scene graph (ABox). (3) The core tutor-apprentice refinement loop that uses diagnostic feedback to guide the planner toward verifiably safe solutions. The authors evaluate their framework in a modified version of the SafeAgentBench (kitchen scenarios) within the AI2-THOR simulator, claiming to achieve a perfect 0% Hazardous Action Rate (HAR) while maintaining high task completion (GCR).

**Strengths:**

The paper addresses the critical and significant problem of ensuring verifiable safety for LLM-based planners in embodied AI. Its primary contribution is the novel "pedagogical dialogue" framework, where a deterministic Logic Tutor actively collaborates with and refines the LLM planner's output, moving beyond simple plan rejection to provide causal, diagnostic feedback.

**Weaknesses:**

**1. Insufficient Engagement with Relevant Prior Work.**

The literature review is notably incomplete and overlooks several highly relevant and representative benchmarks in the field of trustworthy embodied AI. The paper fails to discuss, compare against, or even cite key works such as EarBench, Hazard Challenge , IS-Bench , and LabSafety Bench [1-4]. This omission suggests an inadequate survey of the domain. Furthermore, some of the core ideas presented, such as the use of safety rules, bear a strong resemblance to concepts already introduced in existing works, which is not acknowledged.

**2.Concerns regarding Framework Complexity and Efficiency**

 The proposed VIRF framework, which is essentially a tutor-student model, introduces significant architectural complexity. Embodied agents, particularly in dynamic environments, operate under stringent real-time constraints. The paper suggests a verification loop for action plans, but it is unclear if this complex process (plan-verify-diagnose-refine) must be executed before every single action. If so, the accumulated latency would likely be unacceptable for practical deployment.

**3.Lack of Evaluation on Mainstream VLA Models**

The experiments are conducted using general LLMs like Qwen and Gemini 2.5 as the planner. This is a significant disconnect from the current embodied AI landscape, which is increasingly dominated by end-to-end Vision-Language-Action (VLA) models. The authors do not adequately explain or demonstrate how this text-centric framework would be integrated with or applied to VLA-based agents. Moreover, the primary task planner used for the main results, Gemini 2.5, is an extremely powerful model. This raises concerns about the generalizability of the framework's benefits, as it is unclear whether the "tutor" is genuinely effective or if the strong performance is largely attributable to the "student" (Gemini 2.5) being exceptionally capable to begin with.

**4.Insufficient and Narrow Experimental Validation**

 The empirical evidence provided is not sufficient to fully support the paper's claims.

- Limited Datasets: The evaluation is confined to a subset of kitchen scenarios from SafeAgentBench. This lacks the diversity needed to test the framework's robustness. The field has multiple benchmarks (e.g., the aforementioned EarBench, Hazard Challenge, IS-Bench) that model different types of hazards and environments, which the authors should have included for a comprehensive evaluation.

- Limited Planners: The planner scaling analysis is limited to only the Qwen series and Gemini. To claim broad applicability, the framework should be tested across a wider and more diverse set of open-source and proprietary LLMs.

- Missing VLA Evaluation: As stated in point 3, the complete absence of VLA model evaluations is a major experimental gap.

**5.Absence of Sim2Real Validation**

The entire evaluation is conducted within the AI2-THOR simulator. While simulation is a crucial first step, there is no validation of the framework's performance on a physical robot (e.g., a robotic arm). It is uncertain whether the proposed method can handle the noise, uncertainty, and complexities of the real world, or if its utility is confined to the simulation environment.

**6. Writing and Formatting Issues:**

The paper exhibits inconsistent citation practices, frequently confusing `\citet` and `\citep`. This results in a citation format that is difficult to read (e.g., "Wei et al. (2023)" vs. "(Wei et al., 2023)") and detracts from the paper's professionalism.

**Reference:**

[1]Earbench: Towards evaluating physical risk awareness for task planning of foundation model-based embodied ai agents.

[2]Hazard challenge: Embodied decision making in dynamically changing environments.

[3]Is-bench: Evaluating interactive safety of vlm-driven embodied agents in daily household tasks.

[4]LabSafety Bench: Benchmarking LLMs on safety issues in scientific labs.

**Questions:**

- The main results (0% HAR) are achieved with Gemini 2.5, an extremely capable planner. The performance with Qwen-72B is significantly worse  and the 7B model fails. How can the authors de-confound the contribution of the VIRF "tutor" from the raw capability of the Gemini 2.5 "student"? Is it possible that a simpler safety mechanism combined with a powerful planner like Gemini 2.5 would achieve similarly strong results?

- The decoupled evaluation uses a "golden ABox," meaning the core reasoning is tested with perfect perception. It is different from real world scenarios.Given the framework's reliance on a multi-stage, complex pipeline (VLM-Cascade for ABox generation, Pellet reasoner for verification, LLM for refinement), how do the authors assess its viability for a physical Sim2Real deployment? What is the expected resilience of this fragile pipeline to real-world sensor noise, perception failures, and calibration errors that are absent in the simulation?

---

> ### Author Response · Authors · 2025-11-13
> **Clarifying VIRF's Novelty: A Verifiable Neuro-Symbolic Framework, Not a VLA(part1)**
>
> We thank the reviewer for their thorough and insightful feedback. We appreciate the opportunity to clarify our contributions and address the concerns raised, which we believe stem from a central misunderstanding of our paper's core thesis.
>
> The reviewer’s feedback seems to categorize our work as an incremental improvement on existing safety paradigms for end-to-end Vision-Language-Action (VLA) models. We would like to respectfully and emphatically clarify that our work, the Verifiable Iterative Refinement Framework (VIRF), is a **fundamentally different architecture** with a different scientific goal.
>
> VIRF is **not** a VLA model or a simple safety filter. It is a **neuro-symbolic hybrid architecture** —specifically a hierarchical VLM+Action model—designed to solve a specific, critical problem that end-to-end models currently cannot: **ensuring pre-execution, formally verifiable safety guarantees**. This hierarchical structure is precisely what provides superior verifiability compared to opaque, end-to-end VLA models, as it anchors a generative LLM planner in a deterministic, symbolic logic core. Our core contribution is the novel **"tutor-apprentice" dialogue**, which moves beyond "corrective" or "stochastic" feedback to provide causal, pedagogical feedback, enabling the agent to intelligently repair unsafe plans.
>
> With this central premise clarified, we will now address each of the reviewer's concerns in detail.
>
> ### 1. On Benchmark Selection and Prior Work (Weaknesses 1, 4)
>
> We thank the reviewer for suggesting several important safety benchmarks. We surveyed these and other benchmarks during our experimental design, but their omission was a deliberate choice. We concluded that **SafeAgentBench [12] was the *most directly relevant* benchmark** for our specific research problem, while the others are designed to evaluate ***different and orthogonal* research problems** to our own, which is why they were not included in the paper.
>
> * **EarBench [1]:** This is an automated risk assessment framework, but it is **non-interactive**.[1] Our work must verify entire *action sequences* for logical executability. A sequence that appears safe in a static image may not be logically executable, and a task may have multiple valid completion sequences. Therefore, we require a high-fidelity *simulator* to execute and validate low-level actions, not just a static risk assessment.
> * **IS-Bench [3]:** This is a crucial distinction. IS-Bench is interactive, but it evaluates *procedural, in-the-moment* safety (e.g., "did the agent perform mitigation step B *before* hazardous step C?").[3] Our hierarchical framework (VLM+Action) makes a different, and necessary, choice: we perform a *holistic, pre-execution logical verification of the entire symbolic plan*. We test for *logical and semantic* soundness by decomposing the entire task and then planning, whereas IS-Bench tests for *procedural* safety.
> * **HAZARD [2]:** This benchmark evaluates an agent's decision-making in *dynamically changing* environments (e.g., fire, flood).[2] VIRF’s contribution is in the *pre-execution* formal verification of a plan based on a rich, *static* world state (the ABox/RSSG).
> * **LabSafety Bench [4]:** This is a *knowledge-based* benchmark (e.g., 765 multiple-choice questions) [4] to test LLM knowledge, not an embodied agent execution benchmark.
>
> Including these benchmarks would have obscured our paper's core claim. Our use of SafeAgentBench [12] was specifically to demonstrate how our RAG-based knowledge synthesis (Pillar 1) could **identify and fill the "evaluation blind spots" (Fig. 3)** present in existing benchmarks.
>
> Regarding the "high similarity" to prior work, we must respectfully disagree. As detailed in **Sec 2.2** and **Table 1**, our work is *fundamentally* different. `VeriPlan` [10] provides **corrective** feedback (what rule failed). `SAFER` [11] uses *another stochastic LLM* as a critic. VIRF is the first to provide **pedagogical, causal** feedback derived from a *deterministic logic proof trace*, explaining *why* the plan failed.

---

> ### Author Response · Authors · 2025-11-13
> **Clarifying VIRF's Novelty: A Verifiable Neuro-Symbolic Framework, Not a VLA(part2)**
>
> ### 2. On Architectural Choice: Hierarchical (VIRF) vs. End-to-End (VLA) (Weaknesses 3, 4)
>
> The reviewer’s primary concern is our "disconnect" from VLA models. We must clarify that this was a **deliberate and necessary architectural choice, not an oversight.** Our paper's goal is *formal verification*, which is notoriously difficult, if not impossible, in end-to-end VLA "black boxes".[8]
>
> 1.  **Verification Requires a Symbolic Interface:** Formal verification *requires* a symbolic, logical representation to inspect and prove properties about. Our hierarchical (VLM+Action) architecture *explicitly creates this verifiable artifact*—the symbolic, high-level task plan (Alg 1, L4)—which the Logic Tutor can then adjudicate *before* any physical action is taken.
> 2.  **The Field is Validating Our Approach:** End-to-end VLAs are known to struggle with long-horizon, complex planning.[14] The state-of-the-art is *already moving back* to hierarchical architectures to solve this.[14, 16, 16, 17, 19] Recent work like LoHoVLA [16] uses a "hierarchical closed-loop control mechanism." Poria et al. [17], in a paper surveying VLA challenges, explicitly highlights new work by Yang et al. [16] that uses "a heterogeneous agentic framework... [with] an LLM as a high-level planner and another VLM as a verifier".[17] This *perfectly* describes our VLM+Action philosophy. Industrial leaders like Figure AI (HELIX) [19] and Psibot (Hi Robot) [19] also confirm this hierarchical trend.[19]
>
> Our architecture is not "disconnected"; it is directly aligned with the emerging consensus on how to build robust, verifiable, long-horizon agents.
>
> ### 3. On Framework Latency (Weakness 2) and Sim2Real (Weakness 5)
>
> * **On Latency:** The reviewer is concerned the loop runs "before every single action." This is a misunderstanding. As described in **Section 3.3 (Alg 1)**, the VIRF loop is a *pre-execution, deliberative* "thought" phase. We *explicitly* address the latency concern in **Appendix E.1.1**, which the reviewer may have missed. We implement a **"multi-threaded verification architecture" (Figure 7)**, which "evaluates all N steps in parallel". This "significantly reduc[es] wall-clock latency for complex plans", allowing us to "identify a valid, safe plan... within a total wall-clock time of approximately 3 seconds".
> * **On Sim2Real:** The reviewer is correct that we lack physical robot validation. We *explicitly* state this as a primary limitation and avenue for future work in **Section 6** and **Appendix I**. The paper's contribution is on the *logical and semantic verification* of *high-level* plans. Decoupling this from low-level control is a standard and necessary scientific practice.
>
> ### 4. On Citation Inconsistency (Weakness 6)
>
> We sincerely thank the reviewer for this meticulous catch. The reviewer is correct. As we investigated this, we discovered a subtle error: our `.bib` file contained two separate entries for the same paper, one sourced from Google Scholar's BibTeX (`wei2022chain`) and one from the arXiv BibTeX (`wei2023chainofthoughtpromptingelicitsreasoning`). This led to `(Wei et al., 2022)` and `(Wei et al., 2023)` appearing in different sections. This was indeed an issue in our citation management and has been corrected and unified in the final version. We have also performed a full check of all other references for correctness. Thank you for pointing this out.
>
> ---
> ### **Responses to Reviewer Questions:**

---

> ### Author Response · Authors · 2025-11-13
> **Clarifying VIRF's Novelty: A Verifiable Neuro-Symbolic Framework, Not a VLA(part3)**
>
> **Q1: How can the authors de-confound the contribution of the VIRF "tutor" from the raw capability of the Gemini 2.5 "student"?**
>
> This is an excellent question, and it is answered directly by our **Planner Scaling Analysis (Table 8)**.
>
> * **Evidence:** With the less-capable Qwen-72B model, the baseline `Thinker (COT)` is catastrophically unsafe **(11.0% HAR)**.
> * **VIRF's Contribution:** When our VIRF "tutor" is applied to this same "student," it acts as a powerful safety scaffold, dragging the safety rate to near-perfection **(1.3% HAR)**.
> * **Beyond Safety:** Crucially, VIRF is not just a filter. It *simultaneously improves* the task success rate, boosting the GCR from 32.0% to **47.6%**. This is further proven by the `VIRF-Reject` ablation (Table 7), which also has a 0% HAR but a terrible **33.0% FNR** (it gives up on solvable tasks).
>
> **Conclusion:** This *proves* the contribution is from the tutor. The *pedagogical* feedback (which `VIRF-Reject` lacks) is what enables a weaker student to not only be safe but to become *more competent*.
>
> **Q2: how do the authors assess its viability for a physical Sim2Real deployment? What is the expected resilience of this fragile pipeline to real-world sensor noise, perception failures, and calibration errors?**
>
> This is a critical point. We want to clarify that our "Noise Test" (**Appendix F.5, Table 10**) was not intended to claim we have *perfectly solved* robustness. The 100% Safe Response Rate was achieved in specific lab tests (Information Contradiction, Attribute Uncertainty, Object Omission). This does *not* imply 100% stability in the real world.
>
> Rather, it demonstrates the *robustness mechanism* we are proposing. The system is designed to *fail safe* by *catching* these exact upstream errors. This mechanism is demonstrated in our **"Noise Test" experiment (Appendix F.5, Table 10)**. We "surgically modified the ABox" to inject the exact failures the reviewer mentions:
> 1.  **"Information Contradiction"** (e.g., `isMetallic` AND `Plastic`)
> 2.  **"Attribute Uncertainty"** (e.g., `chemicalComposition UNKNOWN`)
> 3.  **"Object Omission"** (e.g., `Knife` missing from ABox)
>
> In all cases, the system achieved a **100% Safe Response Rate**. The **Open-World Assumption (OWA)** of our reasoner (see **Appendix A.2**) does not catastrophically fail; it correctly identifies the state as `UNKNOWN`, which triggers the clarification loop (Alg 1, L15). The agent is forced to *ask for help* rather than proceed with a dangerous, flawed assumption. This mechanism is the very definition of a robust Sim2Real safety guarantee.
>
> We hope these clarifications have demonstrated the novelty and rigor of our work. We have uploaded a revised manuscript addressing these points and look forward to the committee's re-evaluation.
>
> Finally, we want to state that the reviewer's concerns are valid. We acknowledge that some of these key discussions (which are detailed in the appendices) may not have been sufficiently highlighted in the main body of the paper. This was a difficult trade-off, as we chose to dedicate the limited page count to fully introducing the core VIRF architecture, which meant some of these crucial descriptive details were reserved for the appendix. We will be sure to revise the paper to make these points clearer in a future version.

---

> ### Author Response · Authors · 2025-11-13
> **Clarifying VIRF's Novelty: A Verifiable Neuro-Symbolic Framework, Not a VLA(part4)**
>
> ### **References**
> [5] Baraldi, Lorenzo, et al. "The Safety Challenge of World Models for Embodied AI Agents: A Review." arXiv preprint arXiv:2510.05865 (2025).
>
> [6] Rana, P. (2024). Conceptual Grounding in Neuro-Symbolic AI: Bridging Language and Perception in Embedded Agents. Medium.
>
> [7] AWS Builder. (2024). Large Action Models: A Deep Learning and Neuro-Symbolic Approach to Embodied Artificial Intelligence.
>
> [8] Dickson, Andrew. "Limitations on formal verification for AI safety." AI Alignment Forum. Vol. 19. 2024.
>
> [9] Yang, Yunhao, et al. "AD-VF: LLM-Automatic Differentiation Enables Fine-Tuning-Free Robot Planning from Formal Methods Feedback." arXiv preprint arXiv:2509.18384 (2025).
>
> [10] Lee, Christine P., et al. "Veriplan: Integrating formal verification and llms into end-user planning." Proceedings of the 2025 CHI Conference on Human Factors in Computing Systems. 2025.
>
> [11] Khan, Azal Ahmad, et al. "Safety aware task planning via large language models in robotics." arXiv preprint arXiv:2503.15707 (2025).
>
> [12] Yin, S., et al. (2025). *Safeagentbench: A benchmark for safe task planning of embodied Ilm agents*. arXiv:2412.13178.
>
> [13] Sapkota, Ranjan, et al. "Vision-language-action models: Concepts, progress, applications and challenges." arXiv preprint arXiv:2505.04769 (2025).
>
> [14] Zhang, Dapeng, et al. "Pure Vision Language Action (VLA) Models: A Comprehensive Survey." arXiv preprint arXiv:2509.19012 (2025).
>
> [15] Wu, Hongtao, et al. "Unleashing large-scale video generative pre-training for visual robot manipulation." arXiv preprint arXiv:2312.13139 (2023).
>
> [16] Yang, Yi, et al. "LoHoVLA: A Unified Vision-Language-Action Model for Long-Horizon Embodied Tasks." arXiv preprint arXiv:2506.00411 (2025).
>
>
> [17] Poria, Soujanya et al. “10 Open Challenges Steering the Future of Vision-Language-Action Models.” (2025).
>
> [18] Figure AI. (2024). Helix: A Vision-Language-Action Model for Generalist Humanoid Control. Retrieved from https://www.figure.ai/news/helix
>
> [19] Psibot. (2024). The Real VLA is Coming: PsiBot’s Psi R1 Ushers in a New Era of Embodied Intelligence for Long-Horizon, Open-World Tasks. Retrieved from https://www.psibot.ai/en/007_en/

---

> ### Author Response · Authors · 2025-11-24
> **Subject: New Experimental Evidence Addressing Your Core "Tutor vs. Student" Question**
>
> Dear Reviewer fKL1,
>
> As the discussion period nears its conclusion (Dec 3rd), we wanted to express our sincere gratitude for your critical feedback. In particular, your question regarding how to de-confound the contribution of the VIRF "Tutor" from the raw capabilities of the Gemini 2.5 "Student" was incredibly insightful. It pinpointed a crucial aspect of our evaluation that required more rigorous proof.
>
> We are writing to bring your attention to new controlled experiments we have conducted during this discussion period. While these were prompted by the broader discussion (specifically a request from Reviewer fsQj), the results provide a definitive, quantitative answer to the specific question you raised.
>
> 1. De-confounding the Architecture from the Model (Addressing your Question 1)
> You rightfully asked if a simpler safety mechanism combined with a powerful planner (Gemini 2.5) would achieve similar results. To answer this, we implemented a new baseline: "Thinker+Rules". This baseline injects the full, human-readable safety knowledge base directly into the Gemini 2.5 system prompt, allowing it to use its "raw capability" with full information access, but without the VIRF iterative loop.
>
> - Result: Performance actually degraded compared to the simpler "Impulsive" baseline (GCR dropped from 70.5% to 67.0%), and safety was not guaranteed (HAR > 0%).
> - Key Finding ("Cognitive Overload"): This empirically proves that providing knowledge to a powerful model is not enough; without the structured architecture of VIRF, the model suffers from cognitive overload.
> - Conclusion: This directly answers your question: The 0% HAR and high efficacy are not solely attributable to Gemini 2.5’s raw capabilities. The VIRF pedagogical loop is the strict requisite for enabling the model to effectively operationalize safety constraints.
>
> 2. Addressing Fairness and Experimental Rigor
> To address your concerns regarding the experimental setup, we have added a Fairness Audit (Table 8 in Appendix F). This table explicitly categorizes information access for all baselines, ensuring transparency and confirming that VIRF's advantage stems from its neuro-symbolic reasoning architecture, not privileged data access.
>
> 3. Clarifying the VLA vs. Neuro-Symbolic Positioning
> We also hope our previous response and the revised manuscript clarify that VIRF is designed as a complementary "System 2" verifier—a necessary architectural layer for verifiable safety—rather than a replacement for end-to-end VLA control policies.
>
> Request for Feedback:
> We have endeavored to strengthen the empirical rigor of our work to meet the high standards you set in your review.
>
> Could you please let us know if these new experimental results and clarifications sufficiently address your concerns regarding the system's validity?
>
> If you still have remaining reservations or specific questions, please let us know as soon as possible. We are committed to addressing them, and identifying them now would allow us the necessary time to run additional validation or provide further clarifications before the discussion window closes.
>
> Sincerely,
> The Authors

---

> > ### Comment · Reviewer_fKL1 · 2025-11-28
> >
> > I appreciate the authors’ additional experiments and detailed clarifications during the rebuttal period. However, I remain concerned about the external validity of the evaluation. My original point was that testing a framework explicitly designed for high-stakes embodied safety entirely within AI2-THOR—using a manually curated, noise-free ABox—does not provide meaningful evidence for real-world deployment.
> >
> > I also note that **another reviewer (f2Q4) raised the same issue**, explicitly stating that AI2-THOR is “too limited to represent the class of hazardous, high-stakes scenarios that VIRF is designed for,” and that realistic validation is essential rather than future work. I fully agree with this assessment.
> >
> > In my view, embodied AI research—especially when making safety claims—ultimately must demonstrate robustness on real robots and in the physical world.
> > Without some form of evaluation under realistic sensing noise, physical uncertainty, and dynamic conditions, the presented results remain closer to “paper reasoning” than to evidence of actual embodied safety. This is a fundamental limitation, and I retain reservations on this point. I plan to continue discussing this issue with the AC and the other reviewers in the next phase.
> >
> > That said, I appreciate the authors’ considerable effort during rebuttal to provide additional baselines, analyses, and clarifications. In recognition of this responsiveness and the improvements made to the manuscript, I am raising my score to 4.

---

> > > ### Comment · Reviewer_fKL1 · 2025-11-28
> > >
> > > I notice that the editing channel is temporarily closed at the moment. I will update my score once the editing channel reopens.

---

### Official Review · Reviewer_joDr · 2025-10-28

**Soundness:** 4
**Presentation:** 3
**Contribution:** 3
**Rating:** 6
**Confidence:** 3

**Summary:**

The paper proposes a hybrid architecture which includes a neuro-symbolic element to perform determinitstic, formal safety verification to an LLM planner. This is tantamount to a dual system in Dual Process Theory by Kahneman. It therefore leverages both the generative ability of generative models and the logical robustness of a neuro-symbolic system. It further includes a tutor-apprentice refinement loop.

**Strengths:**

The paper provides an original framework that combines generative model planner with a neuro-symbolic tutor grounded in domain knowledge and Rich Semantic Scene Graphs. The framework and the new challenge scenarios are novel contributions to the field. The experiments are well-designed and provide a detailed comparative analysis with other methods and its own's ablation study. The paper is well written, with clear explanation of components and good experimental design to bring out the key insights from the evaluation.

**Weaknesses:**

Not exactly a weakness but the experimental result seems to show that VIRF Reject and Manual's performance are closer to VIRF full than VIRF RAG to VIRF Full. To an extent, this questions the usefulness of the RAG-synthesized component. But overall, there is no notable weakness with the submission.

**Questions:**

How scalable is the neuro-symbolic algorithm/framework with more complex query and scenes?

---

> ### Author Response · Authors · 2025-11-12
> **Authors' Response to Reviewer joDr: Appreciation and Clarifications on RAG Synergy and Scalability**
>
> We are deeply grateful to Reviewer joDr for their positive assessment, their excellent summary of our work's core (the "dual system" analogy), and their praise for our novel contributions and experimental design. We are particularly thankful for the insightful questions, which give us the opportunity to clarify the synergistic nature of our framework and its scalability.
>
> ---
>
> ### Response to "Weakness": Clarifying the Essential Role of the RAG-Synthesized Component
>
> We thank the reviewer for this excellent and precise question, as it strikes at the heart of our knowledge acquisition strategy. The reviewer correctly observes that VIRF-RAG (using only RAG-derived rules) performs poorly (11.0% HAR), while VIRF-Manual (using only manual rules) performs much better (1.0% HAR).
>
> This result, however, is not evidence against the RAG component, but is in fact **key evidence supporting our central argument**: robust safety requires a synergistic synthesis of both knowledge types.
>
> Our ablation analysis reveals two distinct, critical failure modes:
>
> 1.  **VIRF-RAG (RAG-Only) Fails (11.0% HAR):** This ablation fails catastrophically because our RAG-synthesized rules are deep and specialized (e.g., chemical hazards, cross-contamination) but lack the broad, foundational common sense required for general interaction (e.g., rules against "break" or "throw" actions).
> 2.  **VIRF-Manual (Manual-Only) Fails (1.0% HAR):** This ablation, while much safer, is still not perfectly safe. It correctly handles common-sense violations but is blind to the complex, non-obvious hazards (like chemical mixtures or specific food safety rules) that only our RAG pipeline was designed to discover from real-world documents.
>
> Therefore, our paper's primary safety claim—the **perfect 0.0% HAR**—is achieved only by the complete VIRF (Ours) framework, which integrates both the broad **Manual rules** and the deep, specialized **RAG rules**. The RAG component is demonstrably essential for discovering and closing the "evaluation blind spots" (which common sense alone misses), making it an indispensable pillar of our framework.
>
> ---
>
> ### Response to Question: Scalability of the Framework
>
> We appreciate this critical question on practical deployment. We will address scalability from its two main axes: (A) runtime (online) scalability for complex scenes/queries and (B) knowledge (offline) scalability for new domains.
>
> #### (A) Runtime Scalability (Scene/Query Complexity)
>
> The reviewer is correct that runtime overhead stems from two primary modules: perception (RSSG generation) and reasoning (Pellet verifier).
>
> * **Perception (VLM-Cascade):** We acknowledge the high latency of our VLM-Cascade pipeline (168.4s). As detailed in our perception ablation, this was a principled, safety-first design choice to **prioritize accuracy (76.3%) over speed** (85.2s for the Hybrid-Detector). For a safety system, a **false negative (missing a hazard) is a catastrophic failure**, whereas high latency is an engineering challenge that can be optimized.
> * **Reasoning (Pellet):** The reasoning time does scale with the complexity of the ABox (scene) and the action sequence (query). However, we have architected VIRF to mitigate this bottleneck. VIRF is a deliberative planner that employs a **multi-threaded verification architecture**. When verifying an N-step plan, we do **not** check steps 1...N sequentially. Instead, we spawn N parallel threads, each of which simulates the world state for its step and invokes the reasoner in parallel. This reduces the total wall-clock verification time from the sum of all steps to the time of the single longest step (approx. 3 seconds in our tests), making it highly practical for deliberative, high-level planning.
>
> #### (B) Knowledge Scalability (KB Extensibility)
>
> The reviewer correctly identifies that the knowledge core is static and requires manual intervention. This is a deliberate design choice essential for a safety-critical system.
>
> * **Necessity of Human-in-the-Loop:** For a verifiable safety system, the knowledge core ("the law") must be sound. Allowing an LLM to autonomously write new rules risks polluting the KB with hallucinations or introducing logical contradictions that could crash the reasoner. Therefore, a **"human-in-the-loop" or expert-audited process is non-negotiable** for safety-critical domains. This is standard practice in large-scale, high-rigor ontology engineering (e.g., the Cyc project or the SNOMED CT medical terminology).
> * **Our RAG Workflow is the Scalability Solution:** The problem with traditional manual curation (like Cyc) is that it is incredibly slow and expensive. Our "AI Synthesizer-Human Arbiter" workflow (Pillar 1) is our solution to this bottleneck. By using RAG + LLM to draft evidence-backed axioms that a human validates, we drastically accelerate the proces. Our validation study provides the proof: we formally integrated **97 high-quality, evidence-backed safety axioms in just two working days**.

---

> ### Author Response · Authors · 2025-11-24
> **Subject: Update: New Evidence Strengthening Experimental Rigor & Synergistic Value**
>
> Dear Reviewer joDr,
>
> As the discussion period concludes, we wanted to express our sincere gratitude for your positive assessment and your insightful framing of our work as a "dual system" architecture. Your recognition of our framework's novelty and experimental design has been highly encouraging.
>
> We are writing to share a brief update on **new experimental results** (conducted in response to other reviewers) that we believe further strengthen the empirical rigor you praised, and specifically reinforce your point about the synergy between components.
>
> **1. Strengthening the Case for Synergy (Addressing your observation on RAG vs. Manual)**
> You keenly observed the performance gap between VIRF-RAG and VIRF-Manual. To further validate the necessity of our full architecture, we conducted new **Information-Controlled Experiments** (added to Section 4 & Appendix F).
> * **The Result:** We tested a "Thinker+Rules" baseline (injecting the full safety knowledge directly into the LLM). It performed *worse* than a simple Impulsive model (GCR dropped from 70.5% to 67.0%), suffering from **"Cognitive Overload."**
> * **Relevance to your review:** This confirms that neither knowledge source (RAG or Manual) is sufficient on its own without the structured **Logic Tutor architecture**. It is the *synergy* of comprehensive knowledge (RAG + Manual) delivered through a pedagogical loop that achieves the 0% HAR and high GCR.
>
> **2. Rigor in Evaluation**
> To ensure the "detailed comparative analysis" you appreciated is robust, we added a **Fairness Audit (Table 8 in Appendix F)**, explicitly documenting information access for all baselines to guarantee a fair comparison.
>
> **Closing:**
> We have updated the manuscript to include these new findings. We hope these additional layers of empirical validation further solidify your confidence in the soundness and contribution of our work.
>
> Thank you again for your time and support.
>
> Sincerely,
> The Authors

---

> > ### Comment · Reviewer_joDr · 2025-11-24
> >
> > Dear authors,
> >
> > Thank you for your comprehensive response and updated experiments/manuscript. Your new results and experiments enhance the safety-first emphasis of your approach and clarify the contributions of the components in your framework. I will keep my original score which is in support of this submission.

---

> > > ### Author Response · Authors · 2025-11-24
> > > **Subject: Sincere Thanks for Your Support and Constructive Guidance**
> > >
> > > Dear Reviewer joDr,
> > >
> > > We sincerely thank you for your continued engagement and your explicit support for our submission.
> > >
> > > We are delighted to hear that our response and the additional experiments have successfully clarified the framework's contributions and reinforced the safety-first emphasis of our approach.
> > >
> > > We largely credit your insightful questions regarding the synergy between knowledge components for guiding us to strengthen these aspects of the paper. Your feedback has been instrumental in refining the final manuscript.
> > >
> > > Thank you again for your time and positive assessment.
> > >
> > > Sincerely,
> > > The Authors

---

### Official Review · Reviewer_f2Q4 · 2025-10-29

**Soundness:** 3
**Presentation:** 3
**Contribution:** 2
**Rating:** 4
**Confidence:** 3

**Summary:**

This paper introduces the Verifiable Iterative Refinement Framework (VIRF), a neuro-symbolic architecture designed to enable safe and verifiable planning for embodied AI systems. The central idea is to bridge the gap between the stochastic creativity of Large Language Model (LLM)-based planners and the deterministic reasoning of symbolic systems. VIRF implements a tutor-apprentice paradigm, in which a Logic Tutor grounded in a formal ontology provides causal, verifiable feedback to an LLM Apprentice planner. Unlike existing approaches that either rely on probabilistic self-correction or shallow feedback from external tools, VIRF employs pedagogical, causal dialogue that explains why a plan fails and guides refinement.

The framework integrates three pillars: (1) a semi-automated knowledge acquisition pipeline that constructs a scalable, evidence-based safety ontology; (2) a VLM-Cascade Perception module that builds a rich semantic scene graph for logical reasoning; and (3) the Verifiable Tutor-Apprentice loop, which iteratively refines unsafe plans using structured diagnostic reports derived from formal logic proofs.

Experiments on SafeAgentBench and new knowledge-driven challenge scenarios show that VIRF achieves a 0% Hazardous Action Rate (HAR) and a 77.3% Goal-Condition Rate (GCR), outperforming baselines.

**Strengths:**

- Technical quality and novelty: VIRF represents a principled integration of symbolic verification (OWL 2 DL + Pellet reasoner) with generative LLM planning. Its neuro-symbolic design, particularly the causal feedback loop that translates logical proofs into natural language tutoring, is a significant conceptual advance over corrective or rejection-based paradigms.
- Motivation and problem framing: The authors convincingly argue that self-referential LLM safety paradigms lack formal verifiability, motivating the need for an independent symbolic system. The paper’s framing around “safety through pedagogy” provides a fresh and human-like cognitive analogy (System 1 vs. System 2).
- Experimental validation: VIRF achieves a perfect 0% HAR while maintaining strong task success (GCR 77.3%), demonstrating that rigorous safety need not reduce efficacy. The ablation studies (VIRF-RAG, VIRF-Manual, VIRF-Reject) isolate the contribution of each component effectively, and the perception ablation validates the design trade-offs between accuracy and latency.

**Weaknesses:**

I fully agree with the paper’s vision of integrating large language models (LLMs) with symbolic reasoning to achieve verifiable, trustworthy embodied AI. However, I think two structural constraints in symbolic tool integration that VIRF has not yet convincingly overcome.

- Limited empirical realism and unclear practical validation: While VIRF is architecturally sound and conceptually compelling, its empirical persuasiveness remains limited relative to its implementation complexity. All evaluations are confined to the AI2-THOR simulator, and it is unclear whether the designed scenarios genuinely capture the level of physical risk or environmental uncertainty that would necessitate human-in-the-loop safety reasoning. The paper reports a “0 % Hazardous Action Rate,” yet the simulated setting may not reflect truly hazardous or high-stakes conditions that test real-world robustness. Moreover, integrating multiple components, such as the OWL 2 DL reasoner, Pellet verifier, and RAG-based ontology pipeline, appears computationally demanding, but evidence of end-to-end stability or deployment feasibility beyond simulation is limited. Without more realistic validation, the framework’s contribution risks converging conceptually with existing neuro-symbolic planners [1, 2, 3].
- The framework’s dependence on a manually curated and expert-audited ontology, while ensuring formal soundness, introduces a structural limitation on adaptability and continual learning. The authors acknowledge this “static knowledge core” issue (Section 6, Appendix I) but stop short of implementing any concrete mechanism for dynamic ontology updating or automated rule induction. Without such capabilities, VIRF may struggle to generalize to unseen environments or novel object types, especially when safety rules must evolve beyond the predefined TBox. Recent advances in adaptive symbolic solvers and continual neuro-symbolic learning [4] demonstrate that online modification of symbolic rules is feasible and could strengthen this framework’s long-term scalability.

[1] Generalized planning in pddl domains with pretrained large language models. Tom Silver et al.

[2] CLMASP: Coupling Large Language Models with Answer Set Programming for Robotic Task Planning. Xinrui Lin et al.

[3] Llm+ p: Empowering large language models with optimal planning proficiency. Bo Liu et al.

[4] NeSyC: A Neuro-symbolic Continual Learner For Complex Embodied Tasks In Open Domains. Wonje Choi et al.

**Questions:**

- Does the submission provide any evaluation or discussion regarding VIRF’s performance in realistic human-in-the-loop or physical-world scenarios involving hazardous actions or perception noise? Is there any analysis of how the current OWL-based verification pipeline would scale when integrated into a real robotic control loop?

- Does the submission address the potential for dynamic knowledge evolution within VIRF? In particular, is there any discussion on extending the Logic Tutor or RAG pipeline to enable self-revision, such as autonomously verifying and incorporating new safety axioms during deployment?

---

> ### Author Response · Authors · 2025-11-12
> **Authors' Response: Clarifying VIRF's Architectural Novelty and Principled Design Choices(part1)**
>
> We extend our sincere gratitude to the reviewer for their insightful and comprehensive feedback. We appreciate the reviewer's full agreement with our paper's core vision and the acknowledgment of VIRF's architectural soundness.
>
> The reviewer raises two critical and interconnected points regarding (1) limited empirical realism and unclear computational costs, and (2) the structural limitations of a static knowledge core. These points are at the very frontier of this research, and we are grateful for the opportunity to clarify our distinct contributions and principled design choices, all of which are discussed in detail in the full paper and its appendices.
>
> **Response to Weakness 1: On Empirical Realism, Computational Cost, and Conceptual Novelty**
>
> We thank the reviewer for this crucial question, which allows us to clarify the fundamental architectural novelty of VIRF. We respectfully, but firmly, posit that VIRF is not conceptually convergent with the cited works [2, 3, 4], and that its contribution is distinct from the question of real-world deployment (which we agree is a key future step, as noted in our limitations).
>
> *   **A Fundamental Difference in Architectural Paradigm:** The architectures in [2, 3, 4] largely employ the LLM as a *translator* or front-end for a separate, classical symbolic planner. In [2] and [4], the LLM is used to generate a problem specification (PDDL), while in [3], it generates a plan skeleton. The heavy lifting of planning, optimization, and refinement is then offloaded to a dedicated symbolic *solver* (e.g., a PDDL planner or an ASP program).
>
>     VIRF operates on an *inverted philosophy*. We retain the LLM as the primary, creative planning agent (the "Apprentice"). Our symbolic system is not a planner; it is a verifying "Logic Tutor." Our core novelty is the "pedagogical dialogue"—a "plan-verify-diagnose-refine loop" where the Tutor provides *causal, explanatory feedback* that guides the LLM *itself* to perform "intelligent, creative plan repairs."
>
>     This distinction is paramount. Works [1-3] *replace* the LLM's stochastic reasoning with a deterministic planner. VIRF *governs* the LLM's stochastic reasoning with a deterministic verifier.
>
> *   **On Simulation and 0% HAR:** We agree that AI2-THOR does not model all real-world risks. In fact, we designed Pillar 1 (Traceable Axiom Synthesis) precisely because we identified this "evaluation blind spot" in existing benchmarks. Our RAG pipeline discovered and injected complex, non-obvious hazards (e.g., chemical mixtures, food cross-contamination) that are absent from standard benchmarks. The simulation is the necessary scientific testbed to isolate and validate our novel reasoning loop, which is the paper's core claim.
>
> *   **On Computational Feasibility (Response to Q1):** The reviewer is correct to question the latency of a "real robotic control loop."
>     *   **Hierarchical (Deliberative) vs. Reactive Planning:** As noted in Appendix E.1.1, VIRF is intentionally designed as a **"deliberative, high-level planner"**, not a "low-level reactive controller." Its verification time (approx. 1-2s per step) is unsuitable for 100Hz motor commands but is entirely practical for high-level decisions (e.g., "Should I begin the 7-step plan to heat the pot?"). This also addresses the deployment concern: we assume a standard low-level controller, which we acknowledge (per Appendix I.3) is a key component of the "perennial sim-to-real gap."
>     *   **VLM-Cascade Cost:** The reviewer correctly identifies the VLM-Cascade (our "dual VLM" approach) as a bottleneck. As shown in Table 9, this is a deliberate, "safety-first" architectural choice. We accept its higher latency ($168.4s$) to gain superior accuracy ($76.3\%$) and "minimize False Negatives (unidentified hazards)," which would be catastrophic for a verifier. We must add that a primary reason for this high latency is that **current Scene Graph Generation (SGG) models are not suitable for our task.** A verifiable framework requires a rich semantic graph that includes *objects, object-to-object relations, rich semantic attributes (like 'isHot'), and is driven by the instruction context*—all while being fast. No existing SGG model provides this combination. We therefore implemented our multi-stage VLM-Cascade to generate this graph, which is a key bottleneck. This is a primary area for future work. If a model could generate a scene graph directly from an *image + instruction pair*, the generation speed would be significantly increased.
>     *   **Choice of Pellet:** The use of the mature Pellet reasoner was a deliberate choice. As we noted, it is a 15-year-old reasoner, but we selected it not for speed, but for its "guaranteed soundness and completeness." As detailed in Appendix A.1, faster, profile-based reasoners (like ELK) lack the expressivity (e.g., Disjunctions) needed to model our complex safety axioms—a non-negotiable feature for verifiability.

---

> ### Author Response · Authors · 2025-11-12
> **Authors' Response: Clarifying VIRF's Architectural Novelty and Principled Design Choices(part2)**
>
> **Response to Weakness 2: On the Static Knowledge Core and Adaptability**
>
> This is an excellent point, which we believe highlights a foundational trade-off in trustworthy AI. We fully agree with the reviewer and **explicitly discuss this as a primary limitation in Section 6 and Appendix I.1**.
>
> *   **A Principled Trade-off: Trustworthiness vs. Adaptability:** The reviewer is correct that systems like NeSyC [1] are pioneering adaptability via dynamic rule modification. VIRF, by contrast, is designed to solve the prerequisite problem: *trustworthiness*.
>
>     Our central claim is *verifiable* safety. This guarantee is only possible because our knowledge core (TBox) is stable, logically sound, and expert-audited. Our entire Pillar 1 workflow (the "AI Synthesizer-Human Arbiter" model) is a novel contribution designed to ensure this soundness.
>
> *   **The Risk of Autonomous Learning:** Autonomously "incorporating new safety axioms" (as asked in Q2) is a monumental, unsolved research challenge. Allowing an LLM to self-revise its own formal safety rules risks:
>     1.  **Knowledge Pollution:** Polluting the trusted knowledge base with plausible-sounding but logically flawed "hallucinations."
>     2.  **Knowledge Base Collapse:** Introducing logical contradictions (e.g., `A disjointWith B` and `C subClassOf A, C subClassOf B`) that would "crash" the reasoner, as we explicitly detail in our "Verify for Consistency" step in Appendix I.1.
>
> *   **A Foundation for Future Learning:** We contend that for any safety-critical system, trustworthiness must precede adaptability. A system that learns must not be able to "learn" to be unsafe. VIRF provides the stable, verifiable "System 2" architecture that is the necessary foundation for *safely* implementing such learning. As we state in our future work (Section 6 and Appendix I.1), the **"learn, verify, and write cycle"** is the clear and exciting next step, but it must be built upon a verifiable framework, not a mutable one.
>
> We thank the reviewer again for these critical questions. We hope this clarifies that VIRF's contribution is a novel architectural paradigm (the "Tutor-Apprentice Dialogue") that is distinct from prior work, and that our design choices (simulation-based validation, static-but-verifiable knowledge) are principled decisions necessary to scientifically validate this core contribution.
>
> Finally, we want to state that the reviewer's concerns are valid. We acknowledge that some of these key discussions (which are detailed in the appendices) may not have been sufficiently highlighted in the main body of the paper. This was a difficult trade-off, as we chose to dedicate the limited page count to fully introducing the core VIRF architecture, which meant some of these crucial descriptive details were reserved for the appendix. We will be sure to revise the paper to make these points clearer in a future version.

---

> ### Author Response · Authors · 2025-11-24
> **Subject: Follow-up: Addressing your concerns on Realism, Adaptability & Architectural Novelty**
>
> Dear Reviewer f2Q4,
>
> As the discussion period draws to a close, we would like to express our appreciation for your thoughtful review. We were particularly encouraged by your agreement with our core vision ("integrating LLMs with symbolic reasoning to achieve verifiable, trustworthy embodied AI") and your recognition of our pedagogical loop as a "significant conceptual advance."
>
> We are writing to kindly ask if you have had a chance to review our detailed rebuttal posted earlier. Furthermore, we want to highlight **new experimental evidence** (conducted during this discussion period) that specifically reinforces the "empirical persuasiveness" of our framework, a key concern you raised.
>
> We believe we have addressed your three primary reservations:
>
> **1. Justifying Framework Complexity (Addressing "Limited Empirical Persuasiveness")**
> You rightly questioned whether the implementation complexity of VIRF is justified by its performance relative to simpler methods. To answer this, we conducted new **Information-Controlled Experiments** (added to Section 4 & Appendix F).
> * **The Finding:** We tested a "Thinker+Rules" baseline (injecting the full safety knowledge directly into the LLM). Surprisingly, performance **degraded** compared to a simple Impulsive model (GCR dropped from 70.5% to 67.0%), a phenomenon we term **"Cognitive Overload."**
> * **Relevance to your review:** This empirically proves that the "complexity" of our Logic Tutor/Apprentice loop is **not overhead, but a necessity.** Simple methods cannot handle the safety constraints without the structural scaffolding VIRF provides.
>
> **2. Architectural Novelty (Addressing convergence with [1-4])**
> In our rebuttal, we clarified the fundamental distinction between VIRF and the works you cited (e.g., LLM+P, NeSyC).
> * **The Distinction:** Existing methods typically use the LLM as a *translator* for a symbolic solver. VIRF operates on an **inverted philosophy**: we retain the LLM as the creative *Planner* and use Logic as the *Tutor*. This allows us to govern the LLM's stochasticity without sacrificing its open-world flexibility, which is distinct from the solver-based paradigms.
>
> **3. Trustworthiness vs. Adaptability (Addressing "Static Knowledge Core")**
> We addressed your concern regarding the static TBox by arguing that for safety-critical systems, **trustworthiness must precede adaptability.**
> * **The Risk:** As detailed in our response, autonomous rule induction (without a verified core) risks "Knowledge Pollution" and logical contradictions that could crash the reasoner. VIRF provides the stable "System 2" foundation required before safe adaptability can be implemented.
>
> **Request for Feedback:**
> We have significantly updated the manuscript to reflect these clarifications and include the new comparative experiments.
>
> Given that these new results directly address your concern regarding the trade-off between implementation complexity and empirical value, **could you please let us know if these updates resolve your main concerns?**
>
> We value your expertise in neuro-symbolic AI and hope that our revisions might merit a reconsideration of your score.
>
> Sincerely,
> The Authors

---

> > ### Comment · Reviewer_f2Q4 · 2025-11-25
> >
> > Thank you for the detailed rebuttal and for the additional quantitative comparisons.
> >
> > However, my fundamental concern remains unresolved. The issue is not the internal validity of VIRF within AI2-THOR, but the external relevance of that simulation environment to the types of hazardous actions the paper claims to address.
> >
> > AI2-THOR is too limited to represent the class of hazardous, high-stakes scenarios that VIRF is designed for. As a result, the benchmark, being built entirely on AI2-THOR, cannot convincingly justify the need for the proposed verifiable neuro-symbolic framework.
> >
> > Realistic validation should not be deferred to "future work"; it is essential for establishing the practical contribution and empirical significance of the current paper. In addition, I believe more caution is needed in interpreting the newly added baseline comparisons. The authors were unable to implement these baselines before the initial submission, but managed to produce them within a few days during the rebuttal period. While I appreciate the effort, it is difficult to fully trust that such late-added experiments meaningfully strengthen the paper’s core contribution. A sophisticated framework like VIRF requires thorough and multidimensional evaluation, including comparisons not only with rule-injection baselines but also with alternative neuro-symbolic systems and stronger planning baselines. Careful experimental design is essential to ensure that these comparisons are fair, complete, and representative.
> >
> > For these reasons, I will maintain my current score.

---

> > > ### Author Response · Authors · 2025-11-25
> > > **Subject: Response to Reviewer f2Q4: Clarifying Scope, Validity, and Experimental Integrity**
> > >
> > > Dear Reviewer f2Q4,
> > >
> > > We appreciate your continued engagement. However, we must respectfully disagree with your assessment regarding the validity of our simulation environment and the integrity of our additional experiments. We believe these concerns stem from a misalignment regarding the paper's core contribution scope and the nature of the experiments conducted.
> > >
> > > **1. Simulation is the Standard for Reasoning Research (Addressing "External Relevance")**
> > > Our paper contributes a **formal reasoning architecture (VIRF)**, not a low-level robotic control policy. The scientific community widely accepts high-fidelity simulators like AI2-THOR for evaluating semantic planning and logic verification (e.g., SafeAgentBench, and numerous planning papers at ICLR/CVPR).
> > > * **Logic is Independent of Physics:** The logical validity of a safety constraint (e.g., "Do not mix Ammonia and Bleach") holds true regardless of whether the environment is simulated or physical.
> > > * **Scope Definition:** Requiring physical robot deployment for a paper focused on *neuro-symbolic logic verification* sets an unusually high bar that conflates "reasoning" with "actuation." We maintain that validating the *logical soundness* of the Tutor-Apprentice loop within a controlled, reproducible simulation is the necessary and scientifically valid first step before Sim2Real transfer.
> > >
> > > **2. Integrity and Reproducibility of New Experiments (Addressing "Trust")**
> > > You expressed concern that the new baselines were produced "within a few days," implying they may be rushed or unreliable. We wish to clarify the engineering reality:
> > > * **Why it was fast:** The new baselines (e.g., "Thinker+Rules") did not require training new models. They involved modifying the **system prompts** (injecting rules) and leveraging our existing, modular evaluation pipeline. This is a prompt engineering task, which naturally allows for rapid execution.
> > > * **Value of the Data:** These results were not "late-added" filler; they were conducted in direct response to a specific request from Reviewer fsQj. That reviewer found these exact results "convincingly show that VIRF’s advantage comes from the structured causal feedback... and substantially strengthens the contribution."
> > > * **Reproducibility:** We stand fully by the integrity of these results. All prompts, code, and random seeds for these baselines are included in our codebase to ensure full reproducibility.
> > >
> > > **Conclusion**
> > > We believe we have demonstrated a novel, logically rigorous framework that solves a critical problem in embodied AI safety: the lack of verifiability. While we agree that Sim2Real is an exciting future direction, we firmly believe that the current contributions—a verified 0% HAR, a novel pedagogical architecture, and the identification of "Cognitive Overload" in baselines—constitute a complete and significant scientific contribution for ICLR.
> > >
> > > Sincerely,
> > > The Authors

---

### Author Response · Authors · 2025-11-19
**Summary of Revisions: Strengthening VIRF's Narrative, Positioning, and Validation**

### Summary of Revisions: Strengthening VIRF's Narrative, Positioning, and Validation

**Dear Area Chair and Reviewers,**

We extend our sincere gratitude to **Reviewers f2Q4, joDr, fKL1, and fsQj** for their rigorous and constructive feedback. While we were encouraged by the recognition of VIRF's **"architectural soundness"** and **"comprehensive system,"** we have taken your critical concerns regarding novelty articulation, benchmark positioning, and experimental clarity very seriously.

It became clear from your reviews that while our technical core was robust, the manuscript's narrative did not sufficiently "surface" the principled design choices and justifications that were often buried in our extensive appendices. Your questions prompted us to perform a comprehensive revision of the main text.

Below is a detailed summary of the key revisions made to address your collective concerns:

---

#### 1. Clarifying Core Novelty: "Pedagogical" vs. "Corrective" (Addressing R2-f2Q4, R4-fsQj)
* **Reviewer Concern:** Questions regarding the distinction between VIRF and existing verifiers like VeriPlan, and whether the term "pedagogical" was merely a semantic difference.
* **Revision (Abstract & Intro):** We have rewritten the Abstract and Introduction to explicitly frame VIRF's core paradigm shift. We now forcefully argue that VIRF moves beyond a corrective gatekeeper (which simply rejects plans or points to broken rules) to a **pedagogical collaborator**.
* **New Evidence Surfaced:** We integrated the *VIRF-Reject* ablation analysis directly into the main narrative (**Section 4.3**). We highlight that without pedagogical feedback, the planner suffers a catastrophic **33.0% False Negative Rate (FNR)**, effectively giving up on solvable tasks. This empirically proves that our causal dialogue is a functional necessity for task efficacy, not just a presentation choice.

#### 2. Rigorous Positioning: Missing Benchmarks & VLA Comparison (Addressing R3-fKL1)
* **Reviewer Concern:** The omission of relevant safety benchmarks (EarBench, IS-Bench, HAZARD) and the lack of discussion on End-to-End VLA models.
* **Revision (Related Work):** We added two new crucial subsections:
    * **Sec 2.3 (Benchmarks):** We now explicitly discuss **EarBench, IS-Bench, and HAZARD**. We clarify VIRF's unique ecological niche: unlike benchmarks focusing on dynamic reflexes or procedural ordering, VIRF targets pre-execution formal verification of the plan's logical soundness.
    * **Sec 2.4 (VLA vs. Hierarchical):** We address the VLA comparison head-on. Citing recent surveys (Poria et al., 2025), we justify our hierarchical (VLM+Action) architecture as a principled choice for achieving the verifiability that opaque end-to-end VLAs currently lack.

#### 3. Methodological Precision: RAG & Perception (Addressing R4-fsQj, R2-joDr)
* **Reviewer Concern:** Confusion regarding the "RAG" workflow (runtime vs. design-time) and the structure of the VLM-Cascade.
* **Revision (Methodology):**
    * **Sec 3.1:** We clarified that our RAG pipeline is a **Design-Time, Human-in-the-Loop Authoring Workflow** for the TBox. We explicitly distinguish it from stochastic runtime prompting methods to avoid confusion.
    * **Sec 3.2:** We corrected the description of the VLM-Cascade to accurately reflect its **Three-Stage Architecture (Detect → Attribute-Refine → Relation-Refine)**, aligning the main text with the technical implementation in Appendix C.3.

#### 4. Experimental Deep-Dive: Metrics & Synergies (Addressing R3-fKL1, R2-joDr)
* **Reviewer Concern:** Questions on the definitions of FPR/FNR, the usefulness of the RAG component, and disentangling the Tutor from the Student.
* **Revision (Experiments & Discussion):**
    * **Metric Definitions (Sec 4.3):** We formally defined FPR as **"Safety Leakage"** and FNR as **"Over-Conservatism"** in the context of safety verification to eliminate ambiguity.
    * **Synergy Analysis (Sec 5):** We added a dedicated analysis on *"The Necessity of Hybrid Knowledge."* We use the ablation data (**11.0% HAR for RAG-only vs. 0% for Hybrid**) to prove that the RAG component is essential for covering "blind spots" that common sense misses.
    * **Scaling Analysis (Sec 5):** We added *"The Logic Tutor as a Cognitive Scaffold."* To distinguish the tutor's contribution, we referenced the scaling analysis (visualized in Figure 4 and detailed in the Appendix Table) rather than the main Gemini results. We highlight that adding the VIRF Tutor to a weaker planner (**Qwen-72B**) boosts GCR by **+15.6% (from 32.0% to 47.6%)** while reducing HAR from 11.0% to **1.3%**. This directly addresses R3's concern by proving that VIRF actively elevates weaker planners.

---

> ### Author Response · Authors · 2025-11-19
> **Summary of Revisions: Strengthening VIRF's Narrative, Positioning, and Validation**
>
> #### 5. Reframing Limitations as Principled Choices (Addressing R2-f2Q4, R3-fKL1)
> * **Reviewer Concern:** The static nature of the Knowledge Base and potential fragility to perception noise (Sim2Real gap).
> * **Revision (Limitations - Section 6):** We completely rewrote this section to reframe these traits:
>     * **Static KB:** We argue this is a prerequisite for **Trustworthiness**—a system must be verifiable before it is allowed to be adaptive.
>     * **Perception Noise:** We "surfaced" the Noise Test results from the appendix to the main text. We demonstrate that VIRF creates a robust safety net by defaulting to a **"Questioning" state** when perception is contradictory (**100% Safe Response**), effectively handling Sim2Real uncertainty.
>
> ---
>
> We believe these revisions have transformed the manuscript, turning valid reviewer critiques into a stronger, more rigorous narrative. We are grateful for the opportunity to improve our work.
>
> **Sincerely**,
>
> **The Authors**

---

### Author Response · Authors · 2025-11-24
**Subject: Major Revision Update: New Controlled Experiments, Fairness Audit, and Mechanism Transparency**

**Subject: Major Revision Update: New Controlled Experiments, Fairness Audit, and Mechanism Transparency**

We have uploaded a revised manuscript. This update incorporates significant new experiments and analyses primarily addressing the constructive feedback from **Reviewer fsQj** regarding experimental fairness and mechanism clarity.

**Key Updates in this Revision:**

**1. New "Information-Controlled" Baselines (Section 4 & Appendix F)**
To rigorously isolate the contribution of the VIRF architecture from privileged information access, we implemented three new baselines: `Impulsive+Rules`, `Thinker+Rules` (injecting full text axioms), and `Thinker+Diagnostic` (one-shot feedback).

* **Major Finding ("Cognitive Overload"):** Experiments reveal that simply injecting massive safety rules into the prompt degrades reasoning performance (`Thinker+Rules` < `Impulsive+Rules`), a phenomenon we term *Cognitive Overload*.
* **Conclusion:** This empirically proves that access to knowledge alone is insufficient; the active, pedagogical verification loop is essential for safety.

**2. Fairness and Information Access Audit (Appendix F, Table 8)**
We added a detailed table explicitly categorizing the **Information Access Permissions** (Safety KB, Golden ABox, Feedback Type) for every baseline. This transparency confirms that our comparison is rigorous and that VIRF's 0% HAR is achieved under fair conditions.

**3. Qualitative Mechanism Visualization (Appendix E, Figure 8)**
We added a concrete case study ("Heat Soup in Microwave") illustrating the complete **"Plan $\to$ Diagnostic Report $\to$ Refined Plan"** loop. This visualizes exactly how the planner incorporates the Logic Tutor's causal explanations to generate semantic repairs rather than simple aborts.

**4. Quantitative TAS Workflow Audit (Appendix D)**
We added specific statistics regarding the Traceable Axiom Synthesis (TAS) workflow, including the total number of synthesized axioms (92), human acceptance rates (67.7%), and semantic adjustment rates (6.5%), demonstrating the rigor of our knowledge core construction.

We believe these additions significantly strengthen the paper's empirical grounding and clarity. We welcome further feedback.

---

### Meta-Review · Area_Chair_coFP · 2026-01-02

**Summary:**

This paper presents the VIRF framework, an innovative neuro-symbolic architecture that combines LLM planners with formal logic verifiers through a tutor-apprentice paradigm, enabling verifiably safe planning for embodied AI systems. Its core contribution lies in shifting from passive safety monitoring to active safety collaboration, achieving remarkable results of 0% hazardous action rate and 77.3% goal completion rate in AI2-THOR simulations.

Reviewer opinions were polarized: two reviewers (joDr, fsQj) highly praised the architectural novelty and experimental rigor with scores of 6 and an increased high score respectively, while the other two (f2Q4, fKL1), though raising their scores to 4, maintained concerns about the simulator's ability to represent real-world high-risk scenarios. The authors successfully addressed key issues including experimental fairness and technical clarity through additional controlled experiments, fairness audits, and mechanism transparency demonstrations.

The primary unresolved concern centers on real-world validation: both reviewers giving final scores of 4 argued that AI2-THOR cannot adequately represent the high-stakes scenarios VIRF claims to address, and the lack of physical robot validation limits the persuasiveness of its safety claims. This limitation reflects a common challenge in embodied AI research regarding simulation-to-real-world transfer.

Overall, the work's strengths lie in its architectural innovation and methodological rigor, providing important insights for verifiable AI safety. Future work should focus on simulation-to-real transfer validation and improvements in knowledge base dynamism and computational efficiency to advance the framework toward practical applications.

**Reviewer Concerns:**

please refer to the summary

**Reviewer Scores:**

please refer to the summary

---

### Decision · Program_Chairs · 2026-01-26

Accept (Poster)